# On 'Economies of Sequence' in the Architectural Theory of Agglomeration: A Case of the Kyoto Tourism Industry

**Akifumi Kuchiki**

College of Bioresource Sciences, Department of International Development Studies, Nihon University, Fujisawa 252-0880, Japan; kuchiki.akifumi@nihon-u.ac.jp

**Abstract:** This paper focuses on an architecture-based theory of agglomeration. An agglomeration is composed of a number of segments such as physical infrastructure facilitation including airports and stations. 'Economies of sequence' can be defined as the sequencing of the segments toward the efficient building of an agglomeration. The main three findings are as follows: first, foreign nationals from Kansai International Airport Granger-cause the number of tourists to Kyoto Prefecture, foreign nationals from Kansai International Airport Granger-cause the number of foreign tourists to Kyoto City, and passengers at JR Kyoto Station Granger-cause the number of tourists to Kyoto City; second, the number of foreign tourists Granger-causes revenues in the tourism industry in the cases of Malaysia, the United States of America, and China; and third, the promotion of the tourism industry to redevelop Kyoto station mall in 1997, the "Kyoto Winter Special" campaign in 2003, and the global "Travel & Leisure" destination SNS (social networking service) promotion in 2015 were effective in building the segments of a tourist agglomeration. In conclusion, the economies of sequence of the segments of a tourism industry agglomeration are as follows: the first priority is to reduce transportation costs for tourists through the renovation of airports and stations, and the next segments for enhancing the level of cultural elements are a branding strategy through the promotion of the tourist industry by campaigns and projects using social network systems.

**Keywords:** tourism industry; agglomeration; architecture theory; segment; economies of sequence; transportation costs

---

## 1. Introduction

Many countries including Japan now face a serious issue in terms of regional development due to the decentralization of the government (see METI 2020). Industrial agglomeration policy has been used as the main promoter of regional development in Asia since the Plaza Accord was agreed in 1985 (see Kuchiki and Tsuji 2008), with such agglomeration defined as the spatial concentration of economic activity.

The organization of an industrial agglomeration involves three factors: the location of the industrial agglomeration, the sequence in which the segments of the industrial agglomeration are built, and the organizational management of the industrial agglomeration. Kuchiki (2019) proposed and classified the various theories on industrial agglomeration into three categories: (i) location-, (ii) architecture-, and (iii) management-based theories—as presented in Table 1.

**Table 1.** Three theories on industrial agglomeration.

| Theory | (i) New Economic Geography | (ii) Flowchart Approach | (iii) Diamond Model |
|---|---|---|---|
| | Fujita et al. (1999) | Kuchiki et al. (2017b) | Porter (1990) |
| **Theory** | **(i) Theory of location** | **(ii) Theory of architecture** | **(iii) Theory of geographical management** |
| Characteristics | To examine the role of location in economic decision-making | Introduction of dynamic processes in segment building | Finding the factors of "competitive advantages" of a region |
| Key factors | Product differentiation in monopolistic competition, and transport costs | Efficient sequencing in building the segments of industrial agglomeration | Four factors: 1. Demand conditions, 2. Factor conditions, 3. Firm strategy, structure, rivalry, 4. Related and supporting industries |

Source: Kuchiki, A. and I. Ohno.

First, (i) 'new economic geography' is used to examine the role of location in economic decision-making. Economic geography allows for the discussion of land use, or where and why economic activity takes place. (Fujita et al. 1999) surveyed the von Thünen analysis of land use, in which economic activity occurs and cities are located, as one stream of location theory based on the German tradition. They then developed a new theory of economic geography by using the factors of product differentiation in monopolistic competition and transport costs.

Second, (ii) 'the flowchart approach' is an architecture-based theory of architecture for industrial agglomeration. Figure 1 shows the flowchart approach in the case of a manufacturing agglomeration. The theory discusses how the segments of an agglomeration are sequenced for their efficient construction.

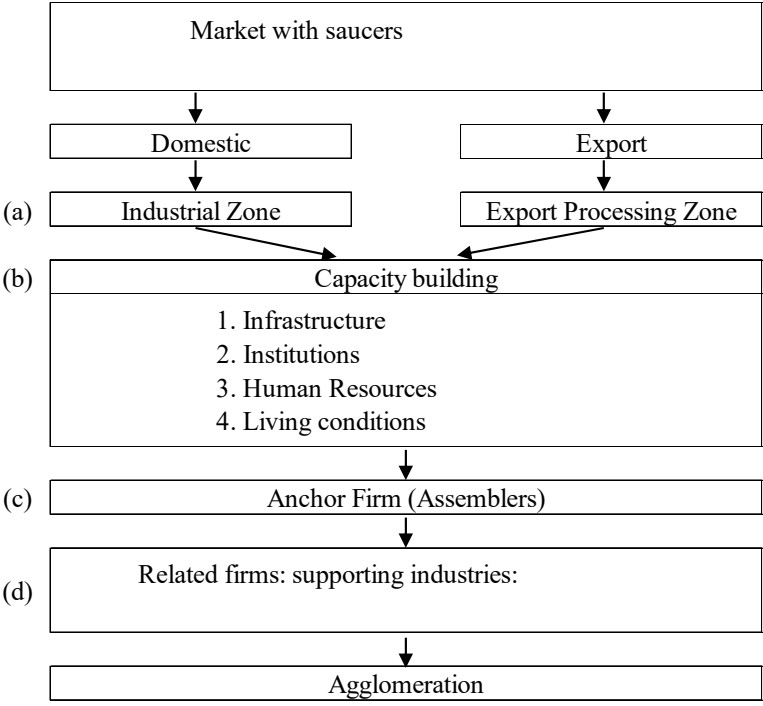

**Figure 1.** The flowchart approach to a manufacturing industry agglomeration. Source: Author.

Third, Porter (1990) developed (iii) his diamond model to specify the four factors, shown in Table 1, contributing to the competitive advantage of a region in terms of geographical management theory.

An agglomeration is composed of a number of segments consisting of physical infrastructure facilitation including ports and highways, institution building, human resources development, and satisfaction with living conditions as is shown in Table 2. A sequence, or iterative order, is the dynamic process through which the segments of an agglomeration are built.

**Table 2.** The segments of an industrial agglomeration.

| Capacity | Segments |
|---|---|
| Infrastructure | Roads<br>Railways<br>Airport<br>Port<br>Communication<br>Water<br>Electricity<br>Industrial zones |
| Human resources | Unskilled labor<br>Engineers<br>Managers |
| Institutions | Laws and regulations<br>Land ownership<br>Foreign exchange system<br>Political system |
| Living conditions | Housing and entertainment<br>Hospitals and schools |
| Living conditions: Cultural aspects | Dynasty: Heian Dynasty<br>Food: Kyoto vegetables<br>Music: Gagaku traditional music<br>History: Temples and shrines<br>Textile: Nishijin textile<br>Handiwork: Kiyomizu ceramic<br>Painting: Water color painting<br>Resort: Amanohashidate<br>Alcoholic beverage: Sake |

Source: Author.

Kuchiki (2019) defined 'economies of sequence' as the selection of any two segments from among the entire group of segments of an industrial agglomeration and the sequencing of the segments toward the efficient building of an agglomeration. The necessary conditions of the economies of sequence hold true if the Granger causality test holds true. The integration of all the economies of sequence into the flowchart approach affords the overall process for efficiently building an industrial agglomeration.

This paper follows the classification of the three theories of (i) location, (ii) architecture, and (iii) geographical management on industrial agglomeration. The paper will neither analyze (i) location nor (iii) geographical management, but focuses instead on (ii) an architecture-based theory of agglomeration. Spatial economics in (i) new economic geography suggests that segments to reduce the transport costs for tourists under a threshold value are the first priority in sequencing segments to satisfy the pre-condition that tourists can move at low costs under a threshold value. The diamond model in (ii) geographical management reasons by analogy that segments to strengthen the competitiveness of a region are the next priority to brand a region from the point of cultural aspects. Accordingly, this paper discusses both segments related to the reduction of transport costs in (i) new economic geography and the segments related to cultural aspects in (ii) geographical management.

Kuchiki and Tsuji (2008) proposed and Kuchiki et al. (2017b) established (ii) a flowchart approach to the architectural theory for building a manufacturing industry agglomeration. There are many papers analyzing the tourism industry, including Yang et al. (2019) and Dibeh et al. (2019), regarding Beijing, China and Lebanon, respectively. However insufficient attention has been paid to tourism industry agglomerations in architectural theory.

The purpose of this paper is to identify the facts of the 'economies of sequence' in tourism industry agglomerations and its attempts to establish an appropriate architectural theory by using Granger

causality tests and a dummy variable method. As an example of the types of industries suited to the formation of industrial agglomerations, this paper will focus on the tourism industry in Kyoto, Japan.

The main three findings are as follows: first, foreign nationals from Kansai International Airport Granger-cause the number of tourists to Kyoto Prefecture, foreign nationals from Kansai International Airport Granger-cause the number of foreign tourists to Kyoto City, and passengers at JR Kyoto Station Granger-cause the number of tourists to Kyoto City; second, the number of foreign tourists Granger-causes revenues in the tourism industry in the cases of Malaysia, the United States of America, and China; and third, the promotion of the tourism industry to redevelop Kyoto station mall in 1997, the "Kyoto Winter Special" campaign in 2003, and the global "Travel & Leisure" destination SNS promotion in 2015 were effective in building the segments of a tourist agglomeration.

In conclusion, the economies of sequence of the segments of a tourism industry agglomeration are as follows: the first priority is to reduce transport costs for tourists through the renovation of airports and stations, the next segments for enhancing the level of cultural elements are a branding strategy through the promotion of the tourist industry by campaigns and projects using social network systems.

In this paper, the sections are organized as follows. Section 2 defines the flowchart approach and concept of 'economies of sequence'. Section 3 focuses on Kyoto in Japan as the prototype of a tourism industry agglomeration. Section 4 finds the facts of the economies of sequence, and Section 5 provides a summary and conclusions.

## 2. The Flowchart Approach and Economies of Sequence

This section establishes the concept of economies of sequence in segment building to formulate the flowchart approach to industrial agglomeration policy through the introduction of a time axis.

(1)　Flowchart approach to transport

Figure 1 provides a summary of the steps in the formation of an agglomeration. The flowchart approach to building a manufacturing industry agglomeration allows the mechanism of an industrial agglomeration to be explained from the perspective of segment building.

In general, the segments consist of physical infrastructure (e.g., roads and electricity supply), institutions (e.g., laws and regulations), and human resources and living conditions (e.g., schools, hospitals, shopping malls, and cultural elements), as presented in Table 2. The segments associated with living conditions, which are key to building the other segments of the industry agglomeration, consist of museums, shopping malls, academic cities, resorts, cultural elements, and so on. The cultural elements themselves consist of historical and artistic elements as well as food, alcoholic beverages, and so on.

The flowchart approach is not purely theoretical. Instead, it provides a practical hypothesis applicable to real-world industrial agglomeration policy as we can effectively form an agglomeration by following the four steps below:

Find ingredients or factors such as industrial zones, means of capacity building, and an anchor firm. Figure 2A shows a, b, c, d, and e.

(i)　Select the minimum number of ingredients from those given above for a flowchart. Figure 2B shows c, a, and e.

(ii)　Order them along a flowchart. Figure 2C prioritizes the ingredients. The number of ways we prioritize them is given by a mathematical 'permutation' as 3! = 3*2*1. In general, n! = n*(n − 1)*(n − 2)* . . . *3*2*1. However, we can only implement one policy. So, we must prioritize policy measures.

(iii)　Specify actors such as central government, local government, non-governmental organizations, or private firms and move forward one step on the flowchart if the answer is 'No'.

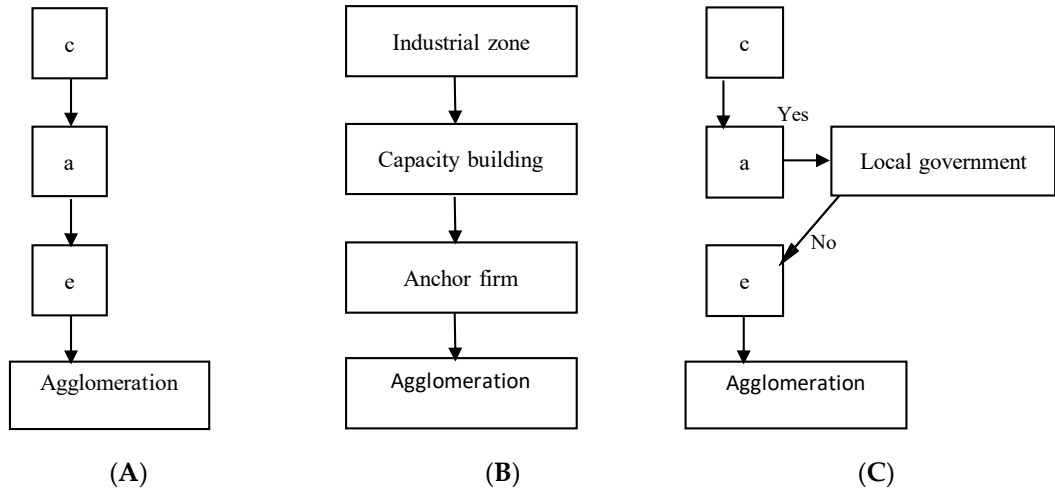

**Figure 2.** The flowchart approach. (**A**): a model; (**B**): an example; (**C**) roles of actors.

The flowchart approach is shown in Figure 3. As mentioned above, the flowchart approach is a practical hypothesis applicable to real-world industrial agglomeration policy because we can form an industrial agglomeration as follows: find segments such as industrial zones, ports, highways; select the minimum number of segments; and then sequence them based on the results for 'the economies of sequence' described in the previous section. The flowchart approach illustrates the dynamic process of segment building in an industrial agglomeration after arranging the segments through the incorporation of the results of economies of sequence. Each step of the flowchart works toward finding economic agents and securing funding for the building of the respective segment.

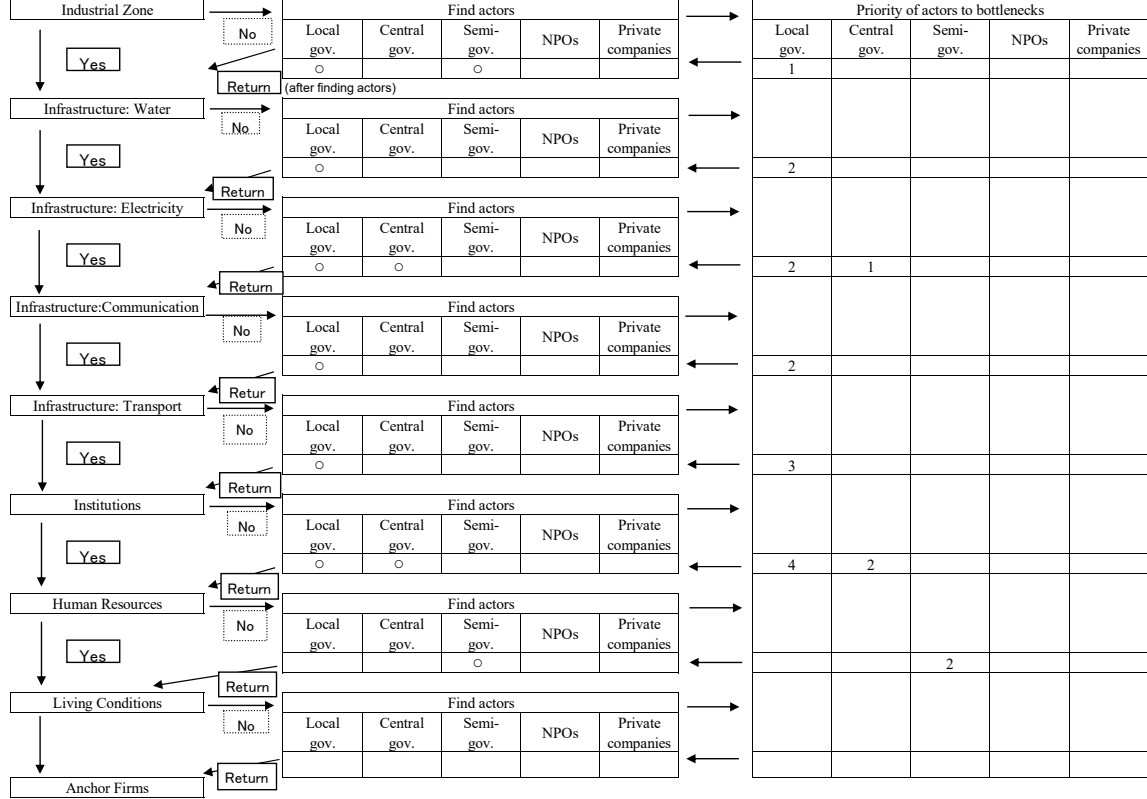

**Figure 3.** The process of finding actors. Source: A. Kuchiki and J. Kabir.

First, we need to identify whether industrial zones have been established. If not, we must decide which actors should establish such zones. We then specify the actors, such as the central government, local government, non-governmental organizations, or private firms, and move one step forward on the flowchart if the answer is 'No'. Once we identify these actors, we return to the mainstream of the flowchart.

We next apply the flowchart's second step, segment building, which takes place after the establishment of the industrial zones. We examine whether the water supply is enough for the industrial zones. We then proceed down the flowchart to examine power supply, communication, and transportation. Figure 4 summarizes the priorities regarding the actors in segment building.

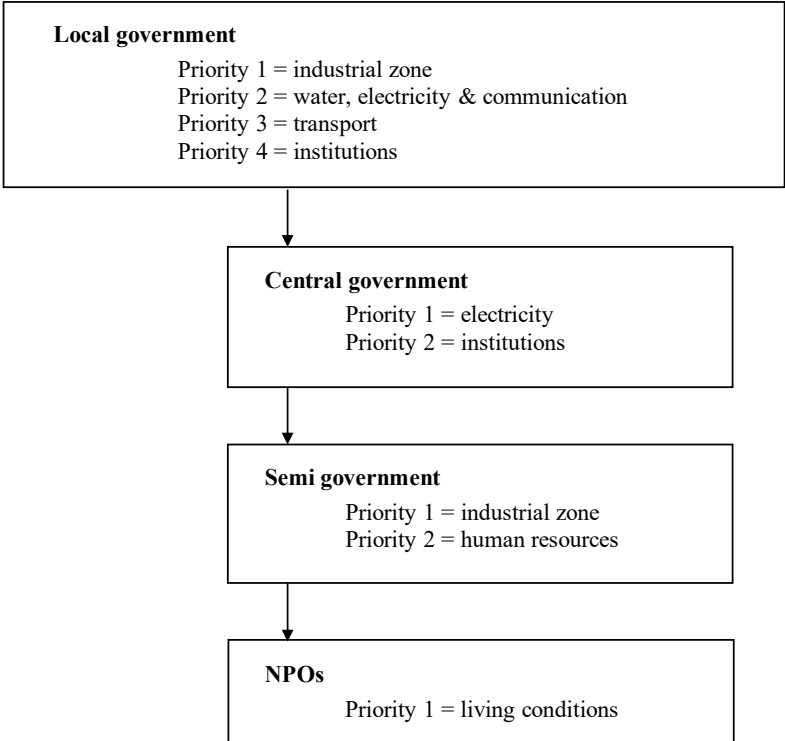

**Figure 4.** The sequencing of each actor. Source: A. Kuchiki and J. Kabir.

(2) Economies of sequence

The flowchart approach is a theory of architecture that allows the efficient sequencing of segments in an industrial organization. The results of 'the economies of sequence' are key to the success in completing an agglomeration and integrated into the flowchart approach to agglomeration policy. This section shows that the necessary conditions of the economies of sequence defined below hold true if the Granger causality test holds true.

The economies of sequence can be mathematically defined as follows. Suppose that there are three periods: the first, second, and third periods. Let us examine two examples of segment formation sequencing in an industrial agglomeration. Policy measures, for which there are two candidate sequences (A and B), form a segment. Suppose that an agglomeration consists of three segments ($s_1$, $s_2$, $s_3$), say a port, a highway, and an industrial zone in the manufacturing sector model.

The difference between A and B lies in the ordering of $s_2$ and $s_3$. In A, the segment formation sequence is assumed to be $s_1$, $s_2$, and $s_3$; thus, it is assumed that the sequence of policy measures to form a production function in B is $s_1$, $s_3$, and $s_2$. Accordingly, A and B can be notated as follows:

$$A = \{s_1, s_2, s_3\}, \text{ and}$$

$$B = \{s_1, s_3, s_2\} = \{\text{a port, an industrial zone, and a highway}\}.$$

The production functions for sequence A and sequence B for gross regional products can be given as $Y_A$ and $Y_B$, respectively. We assume, without loss of generality, that the production functions of sequences A and B are constant returns to scale. The production function after the implementation of the successful policy measures for $\{s_1, s_2, s_3\}$ is $Y_A = f(\{s_1, s_2, s_3\})$. $C_A$ and $C_B$ are the total costs of sequence A and sequence B, respectively, and $c_i (s_k: V_0)$ ($k = 1, 2, 3$; $i = 1, 2, 3$) is the cost to implement the policy measures to form the segments $s_k$ in period i. Suppose that the production functions in sequences A and B in the third period are $Y_A$ and $Y_B$, respectively, or $Y_A = f(A) = f(\{s_1, s_2, s_3\})$.

The production costs of $Y_A$ are

$$C_A = (1 + r)^2 c_1 (s_1: V_0) + (1 + r) c_2 (s_3: V_0) + c_3 (s_2: V_0),$$

where we assume that the other conditions, or $V_0$, are constant.

Now, the productivity of A and B in the third period can be compared. Suppose that there exists economies of sequence between $s_2$ and $s_3$; that is, $Y_B$ is very small, approaching zero, due to the diseconomies of sequence between $s_3$ and $s_2$, while $Y_A$ is large. The segment formation sequence from $s_3$ to $s_2$ is inefficient in comparison to that of the segment formation sequence from $s_2$ to $s_3$.

In other words, the following three cases are possible:

$Y_A > Y_B$ in the case that $\{s_1, s_2, s_3\}$ is more efficient than $\{s_1, s_3, s_2\}$,
$Y_A < Y_B$ in the case that $\{s_1, s_3, s_2\}$ is more efficient than $\{s_1, s_2, s_3\}$,
$Y_A = Y_B$ in the case that $\{s_1, s_3, s_2\}$ is as efficient as $\{s_1, s_3, s_2\}$.

Figure 5 shows the case in which Segment 5 = Segment 1 = Segment 6 = Segment 7. It can be defined as follows.

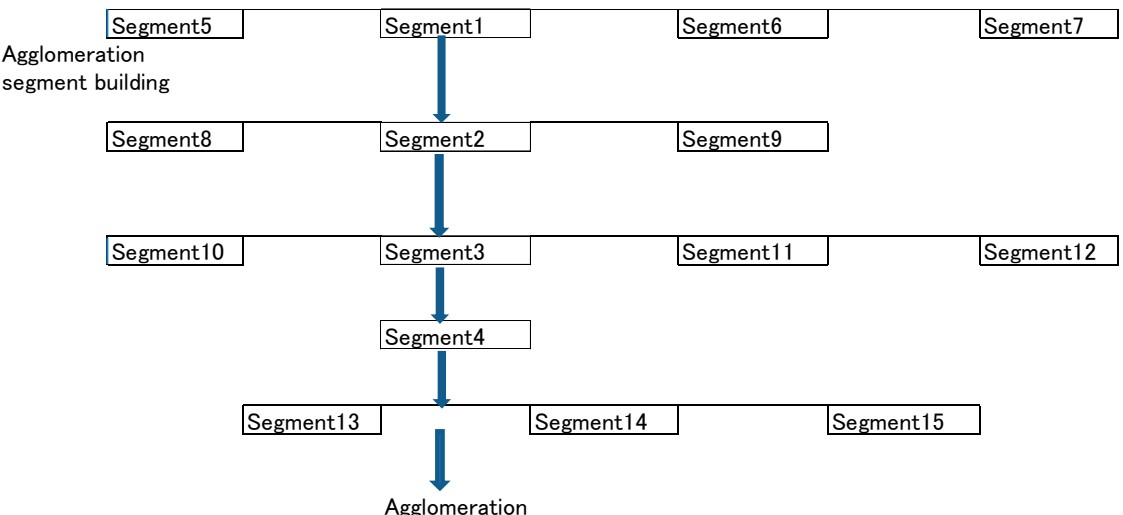

**Figure 5.** Flowchart approach to segment building. Source: Author.

A weak definition of the 'diseconomies of sequence':

$$C_B \rightarrow \infty, \text{ or } Y_B \rightarrow 0, \text{ or } Y_B/C_B \rightarrow 0.$$

We can, thus, define 'economies of sequence' as not 'diseconomies of sequence.' In the case of 'economies of sequence' between $s_2$ and $s_3$,

$$C_A < \infty, \text{ and } Y_A > 0,$$

and the economies of sequence exist between $s_2$ and $s_3$, so that

$$0 < Y_B/C_B < Y_A/C_A.$$

This demonstrates that sequence A is more efficient than sequence B.

This section explains the Granger causality test models used in the paper. In the Granger causality relationship between segment x and segment y, a null hypothesis that segment x does not Granger-cause segment y is rejected.

We suppose that one period of the sequence is two years and that the lag of x and y is 2. Model 1 is an autoregressive model of y. Model 2 is an autoregressive model of x in addition to y. Granger causality is defined as the existence of Granger causality if the prediction error value of Model 2 is smaller than that of Model 1. We suppose that the period of the sequence from one segment to the next segment is two years. That is, the lag of variables is two years.

The equations of Model 1 and Model 2 are

$$y\,(t) = b_{11}\,y(t-1) + b_{12}\,y(t-2) + e_1 \tag{1}$$

$$y\,(t) = b_{21}\,y(t-1) + b_{22}\,y(t-2) + c_1\,x(t-1) + c_2\,x(t-2) + e_2 \tag{2}$$

where the lag of y is 2, a is a constant and $e_i$ (i = 1, 2) is the error terms. Model 1 is regressed using the OLS and its sum of squared residual is noted as $SSR_0$. Similarly, Model 2 is regressed using the OLS and its sum of squared residuals is noted as $SSR_1$.

In this section, Equation (2) is changed into other variations as follows:

$$y\,(t) = b_{21}\,y(t-1) + b_{22}\,y(t-2) + \ldots + b_{2n}\,y(t-n) + c_1\,x(t-1) + c_2\,x(t-2) + \ldots + c_2\,x(t-n) + e_2 \tag{3}$$

where n = 1, 2, 3, 4, or 5.

Analyses in a time series assume stationary stochastic processes. Drift is an intercept component in a time series. Ordinary least square estimations with a drift term (constant term) are non-stationary stochastic processes. It is generally accepted that equations without a drift term are used for stationary stochastic processes; thus, our models adopt Equations (1) and (2) without a drift term after confirmation that equations with a drift term provided results for non-stationary stochastic processes.

In the case that all $c_i$ are zero, x does not Granger-cause y, the null hypothesis is

$H_0$: all $c_i$ = 0.

The null hypothesis holds if

$H_1$: at least one $c_i \neq 0$.

Then x Granger-causes y.

In other words, in the case that the variance of Model 2 is equal to that of Model 1, or

$\sigma\,(Y|U) = \sigma\,(Y|U\text{-}X_{-1})$,

x does not Granger-cause y.

In the case that the variance of Model 2 is greater than that of Model 1, or,

$\sigma\,(Y|U) < \sigma\,(Y|U\text{-}X_{-1})$

x Granger-causes y.

We denote the error sums of squares Model 1 and Model 2 by $SSE_R$ and $SSE_U$.

F value is

$$F = ((SSE_R - SSE_U)/p)/(SSE_U/(n-2p)).$$

Here the numerator is the value of $SSE_R$ minus $SSE_U$ divided by degree of freedom. Its degree of freedom is p, that is, the value obtained by the degree of freedom of $SSE_R$ minus that of $SSE_U$. The value of this case is 2. The denominator is $SSE_U$, or n − 2p.

(3)　　Implications of the new economic geography for the flowchart approach

New economic geography determines the sequence of segments in an industrial agglomeration when the flowchart approach is applied. Sato et al. (2011) conclude that the new economic geography provides the necessary conditions for an industrial agglomeration between two locations as follows: One industrial agglomeration shifts another industrial agglomeration if the transport costs between the two locations fall below 'the threshold value.' This means that segments given the first priority in building an industrial agglomeration are those related to the reduction of transport costs, such as the development of an airport or a station, new time tables for trains, the construction of highways and ports, exemption from import tariff rates, and so on.

The isolated town, or "monocentric city model," of Alonso (1964) reinterpreted the von Thünen model by substituting commuters for farmers. The model of a disc-shaped central business district and surrounding residential region serves as a starting point for urban economic analysis. The work of Kuchiki et al. (2017a), on the 'railway-led formation of the agriculture-food-tourism industry cluster,' developed a model to clarify the impact of new railways on relocating the agriculture sector and manufacturing sector through an examination of the Hankyu Railway in Japan. It showed that the reduction of transport costs makes positive effects on the sustainability of disc-shaped central business district. This paper applies its conclusions to the analysis of the location of population and tourism.

## 3. The Segments of a Station and Cultural Elements in Kyoto

Kyoto has around 1200 years of history, with a large number of World Cultural Heritage sites in the area. Seventeen temples located in Kyoto City, Uji City, and Otsu City are registered as World Heritage sites. Agricultural products for which Kyoto is famed include vegetables, Uji tea, soybeans, and azuki beans. Kyoto produces many characteristic dishes, including pickles and confectionary, and enjoys beautiful natural scenery to be enjoyed in each season. Taken together, this makes Kyoto one of the most original and attractive international tourist destinations.

Regarding the railway transportation segment, the sequencing of the renewal of the station mall followed by the opening of a new transportation line was shown to be efficient based on the economies of sequence. The total floor space of the 60-meter tall Kyoto Station is 236,000 square meters. Various facilities, such as restaurants, hotels, department stores, shopping malls, museums, and theaters, are located at the Kyoto Station mall (see Kyoto Station Building 2017). The project to redevelop Kyoto Station mall began in 1993 and the mall was reopened in 1997, with the project winning a Brunel Award Commendation in 2001. The Kyoto Theater reopened in 2002, and the Kyoto Ramen Noodle Street was newly opened in 2003. The Kyoto General Information Center on Tourism opened in 2010 and a project for the development of the station circle at the south exit of Kyoto Station was finished in 2016 (see Kuchiki et al. 2017a).

The Kansai Airport Express Haruka Line links Kyoto Station and the Kansai International Airport, taking eighty minutes for the trip. The line has become the main means of transportation since the Kyoto City Air Terminal bus services were discontinued in 2002.

Regarding the cultural aspects segment shown in Table 2, Kyoto set a target of 50 million tourists in 2000, and the following four projects were promoted to realize this vision (see The Mayor of Kyoto 2009). First, a platform composed of six groups was created to implement the "Kyoto Flower Paper Lantern" project; second, the "Kyoto Winter Special" project attracted tourists from abroad; third, a project promoting domestic school trips to Kyoto was put in place; and fourth, a project was implemented to cash in on the Kyoto boom in Japan and the Japan boom in Europe and America.

The "Kyoto Flower Paper Lantern" project, which began from March in 2003, sought to create a new tourist attraction with a "lighting" theme. The "Arashiyama Flower Paper Lantern" attraction thereafter started in December 2005. Projects for guests from abroad involved close cooperation with the "Visit Japan Campaign," which was initiated in 2003 by the Japanese government, and the "Kyoto Winter Special" campaign. As a result, the number of guests visiting Kyoto increased from 384,000 in 2001 to 937,000 in 2008 (see The Mayor of Kyoto 2009).

Next, the "Future Kyoto Tourism Promotion Plan 2010" has strengthened the "cultural aspects" since its implementation in fiscal year 2010. Kyoto's website has been providing information for guests from abroad in 8 languages since the 2012 fiscal year, while information on restaurants for Muslim guests has also been provided exclusively on the website since the 2013 fiscal year. In particular, one of the most effective factors for the rapid increase in foreigners is that Kyoto City was ranked first as a global "Travel & Leisure" destination in both 2014 and 2015 by an American travel magazine.

The effects of the construction of the segments explained above related to cultural aspects on the number of tourists can be seen as follows: The number of tourists to Kyoto exceeded 50 million people in 2008 and reached 56.84 million people in 2015, the number of tourists staying overnight reached 13.62 million people in 2015, and the number of foreigners staying overnight also increased markedly from 1.13 million in 2013 and 1.83 million in 2014 to 3.16 million in 2015 (see Kyoto City 2016).

## 4. Methodology and Empirical Results

(1)   The dummy variable method: the importance of a criterion year and interception dummy

Table 3 shows data on the tourism industry of Kyoto to use for Granger causality tests, and Table 4 shows the results for the dummy variable method. Estimated equations can be described as follows:

$$Y_i = a_1 + a_2 T_i + a_3 D_{m,20j} + e_i$$
$$(m = 1, 2, \ldots, 16)$$

where $Y_i$, $T_i$, $D_{m,20j}$, $a_p$ (p = 1, 2, 3), and $e_i$ denote foreign tourists to Kyoto City, year, dummy variables, coefficients, and an error term, respectively.

**Table 3.** Data on tourists and passengers (A, B, C, D: Unit: 10,000).

| Year | Kyoto Pre. Tourists A | Kyoto Sta. JR Passengers B | Kyoto City Tourists C | Kyoto City Foreign Guests D | Kansai Airport Foreign Nationals E |
|---|---|---|---|---|---|
| 2000 | 6286 | 6012 | 4051 | 39.8 | 1,194,740 |
| 2001 | 6396 | 6110 | 4132 | 38.4 | 1,198,460 |
| 2002 | 6522 | 6182 | 4217 | 48.1 | 1,177,532 |
| 2003 | 6703 | 6282 | 4374 | 45 | 1,112,229 |
| 2004 | 6866 | 6348 | 4554 | 54.4 | 1,289,109 |
| 2005 | 7088 | 6438 | 5021 | 73 | 1,369,514 |
| 2006 | 7260 | 6539 | 4839 | 80.3 | 1,505,025 |
| 2007 | 7462 | 6603 | 4727 | 92.7 | 1,662,378 |
| 2008 | 7799 | 6662 | 5021 | 93.7 | 1,652,085 |
| 2009 | 7408 | 6565 | 4690 | 78.4 | 1,357,556 |
| 2010 | 7674 | 6705 | 4955 | 98.4 | 1,751,906 |
| 2013 | 7787 | 7118 | 5162 | 113 | 2,326,263 |
| 2014 | 8375 | 7080 | 5564 | 183 | 3,173,759 |
| 2015 | 8748 | 7321 | 5684 | 316 | 5,012,402 |
| 2016 | 8741 | 7315 | 5524 | 318 | 6,090,300 |
| 2017 | 8687 | 7420 | 5362 | 353 | 7,162,310 |

Source: Author based on A: https://www.pref.kyoto.jp/kanko/research/documents/hyo10.pdf; B: The website of Kyoto Prefecture http://www.pref.kyoto.jp/tokei/yearly/tokeisyo/ts2016/tokeisyo201610.html; https://www.westjr.co.jp/fan/ranking/ (2017). C, D: Kyoto City Official Website. https://www.pref.kyoto.jp/kanko/research/documents/06hyou2.pdf; http://www.pref.kyoto.jp/tokei/; E: Osaka Immigration Bureau Kansai Airport Branch, Kansai Airport Review 2018. Jan., p. 41). http://www.kar.or.jp/wpcms/wp-content/uploads/2018/02/review1802.pdf.

**Table 4.** Dummy variables method (*p*-value).

| | | | | | | | | | | Y$_i$ | T$_i$ | D$_1$ |
|---|---|---|---|---|---|---|---|---|---|---|---|---|
| | | | | | | | | | | 39.6 | 1996 | 0 |
| | | | | | | | | | | 41.4 | 1997 | 0 |
| | | | | | | | | | | 40 | 1998 | 0 |
| | | | | | | | | | | 39.5 | 1999 | 0 |
| 2000 | D1 | 0.0375 | | | | | | | | 39.8 | 2000 | 1 |
| 2001 | D2 | 0.02794 | | | | | | | | 38.4 | 2001 | 1 |
| 2002 | D3 | 0.0295 | 2002 | D3 | 0.4055 | | | | | 48.1 | 2002 | 1 |
| 2003 | D4 | 0.0325 | 2003 | D4 | 0.2126 | | | | | 45 | 2003 | 1 |
| 2004 | D5 | 0.0618 | 2004 | D5 | 0.157 | 2004 | D5 | 0.4634 | | 54.4 | 2004 | 1 |
| 2005 | D6 | 0.1236 | 2005 | D6 | 0.1211 | 2005 | D6 | 0.2882 | | 73 | 2005 | 1 |
| 2006 | D7 | 0.1978 | 2006 | D7 | 0.0747 | 2006 | D7 | 0.1417 | | 80.3 | 2006 | 1 |
| 2007 | D8 | 0.3468 | 2007 | D8 | 0.0582 | 2007 | D8 | 0.0731 | | 92.7 | 2007 | 1 |
| 2008 | D9 | 0.5575 | 2008 | D9 | 0.0540057 | 2008 | D9 | 0.0356922 | | 93.7 | 2008 | 1 |
| 2009 | D10 | 0.9392 | 2009 | D10 | 0.1086 | 2009 | D10 | 0.03943 | | 78.4 | 2009 | 1 |
| 2010 | D11 | 0.4213 | 2010 | D11 | 0.5127 | 2010 | D11 | 0.2231 | | 98.4 | 2010 | 1 |
| 2012 | D12 | 0.1127 | 2012 | D12 | 0.8326 | 2012 | D12 | 0.7013 | | 84.5 | 2012 | 1 |
| 2013 | D13 | 0.0023 | 2013 | D13 | 0.0645 | 2013 | D13 | 0.1726 | | 113 | 2013 | 1 |
| 2014 | D14 | 0.0000079 | 2014 | D14 | 0.0001 | 2014 | D14 | 0.0011 | | 183 | 2014 | 1 |
| 2015 | D15 | 1.77E-10 | 2015 | D15 | 0.0000001 | 2015 | D15 | 0.0000019 | | 316 | 2015 | 1 |
| 2016 | D16 | 0.0000865 | 2016 | D16 | 0.009 | 2016 | D16 | 0.0251 | | 318 | 2016 | 1 |
| | | | | | | | | | | 353 | 2017 | 1 |

Source: Author.

Here the dummy variables are set as follows:

Regarding D$_{m,20j}$,
D$_{m,20j}$ = 0 in the cases of j = 00, 01, 02, . . . , 0m.
D$_{m,20j}$ = 1 in the cases of j = 0(m − 1), 0(m − 2), . . . , 17 (when m + k < 10),
j = (m − 1), (m − 2), . . . , 17 (when m + k > 10).

(The data for 2011 is excluded due to the 2011 earthquake in Japan.)

As is shown in Table 4, the dependent variable of the equationity of Y$_i$ and the equation includes one independent variable 'time' of T$_i$ together with a dummy variable D$_{1,2000}$.

$$Y_i = a_1 + a_2 \, T_i + a_3 \, D_{1,2000} + e_i,$$

where i = 1996, 1997, . . . , 2017. First, the *p*-value is significant in cases in which the starting year was 1996. The null hypothesis is that the dummy variable of 2001 is not significant and the *p*-value of 0.02794 was considered statistically significant. Further, the null hypothesis is that the dummy of 2015 is also not significant and the *p*-value of $1.770949 \times 10^{-10}$ was considered statistically significant.

Second, the *p*-value is significant in cases in which the starting year was 2001. The null hypothesis is that the dummy variable of 2008 is not significant and the *p*-value of 0.05400 was considered statistically significant. Again, the null hypothesis is that the dummy variable of 2015 is not significant and the *p*-value of 0.0000001 was considered statistically significant. The dummy variable is significant and SNS is effective in inviting foreign guests to Kyoto in 2015. The *p*-value is significant in case in which the starting year was 2003 and the estimations confirm that the Visit Japan Campaign was effective in building the tourism agglomeration of Kyoto.

Third, the *p*-value is lower and more significant in cases in which the criterion year was 2001 rather than 2003. The null hypothesis is that the dummy variable of 2008 is not significant and the *p*-value of 0.0356922 was considered statistically significant. Again, the null hypothesis is that its interception dummy of a dummy variable of 2015 alone is not significant and the *p*-value of 0.0000019 was considered statistically significant. This means that the trend over time of the effect of developing Kyoto started in 2001.

Projects aimed at encouraging for guests from abroad involved close cooperation with the "Visit Japan Campaign" started in 2003 by the Japanese government and the internet-based "Kyoto Winter Special" campaign. The projects became effective in 2009.

(2) The Granger causality test: the case of Kyoto

As shown in Table 5, first, a causality test that foreign nationals from Kansai International Airport Granger-causes tourists to Kyoto Prefecture is significant in cases in which the lag is 3 years. Second, a causality test that passengers at JR Kyoto Station Granger-causes foreign nationals from Kansai International Airport is significant in cases in which the lag is 3 years. Third, a causality test that passengers at JR Kyoto Station Granger-causes tourists to Kyoto City is significant in cases in which the lag is 1 year. Fourth, a causality test that foreign nationals from Kansai International Airport foreign guests to Kyoto City is significant in cases in which the lag is 3 years. The above results show that the lag differs in each case.

**Table 5.** Results of Granger causality tests.

| Lag | Segment 1 | Segment 2 | F-Test | *p*-Value | Significance |
|-----|-----------|-----------|--------|-----------|--------------|
| 1 | Foreign nationals from Kansai Airport | Tourists to Kyoto Prefecture | 1.0173 | 0.3225 | |
| 1 | Tourists to Kyoto Prefecture | Foreign nationals from Kansai Airport | 0.64867 | 0.4279 | |
| 2 | Foreign nationals from Kansai Airport | Tourists to Kyoto Prefecture | 1.4894 | 0.2495 | |
| 2 | Tourists to Kyoto Prefecture | Foreign nationals from Kansai Airport | 0.050735 | 0.9507 | |
| 3 | Foreign nationals from Kansai Airport | Tourists to Kyoto Prefecture | 4.5703 | 0.01969 | Yes |
| 3 | Tourists to Kyoto Prefecture | Foreign nationals from Kansai Airport | 0.58198 | 0.6365 | |
| 1 | Foreign nationals from Kansai Airport | Passengers at JR Kyoto Station | 0.096464 | 0.7586 | |
| 1 | Passengers at JR Kyoto Station | Foreign nationals from Kansai Airport | 0.73232 | 0.4 | |
| 2 | Foreign nationals from Kansai Airport | Passengers at JR Kyoto Station | 0.67421 | 0.5208 | |
| 2 | Passengers at JR Kyoto Station | Foreign nationals from Kansai Airport | 0.58996 | 0.5637 | |
| 3 | Foreign nationals from Kansai Airport | Passengers at JR Kyoto Station | 1.9611 | 0.1663 | |
| 3 | Passengers at JR Kyoto Station | Foreign nationals from Kansai Airport | 4.647 | 0.01865 | Yes |
| 1 | Passengers at JR Kyoto Station | Tourists to Kyoto City | 4.918 | 0.03552 | Yes |
| 1 | Tourists to Kyoto City | Passengers at JR Kyoto Station | 0.015575 | 0.9016 | |
| 2 | Passengers at JR Kyoto Station | Tourists to Kyoto City | 2.4624 | 0.1107 | |
| 2 | Tourists to Kyoto City | Passengers at JR Kyoto Station | 0.30084 | 0.7435 | |
| 3 | Passengers at JR Kyoto Station | Tourists to Kyoto City | 1.9902 | 0.1618 | |
| 3 | Tourists to Kyoto City | Passengers at JR Kyoto Station | 0.36771 | 0.7775 | |
| 1 | Foreign guests to Kyoto City | Foreign nationals from Kansai Airport | 1.3005 | 0.2645 | |
| 1 | Foreign nationals from Kansai Airport | Foreign guests to Kyoto City | 0.42631 | 0.5195 | |
| 2 | Foreign guests to Kyoto City | Foreign nationals from Kansai Airport | 0.095063 | 0.9097 | |
| 2 | Foreign nationals from Kansai Airport | Foreign guests to Kyoto City | 0.78873 | 0.4681 | |
| 3 | Foreign guests to Kyoto City | Foreign nationals from Kansai Airport | 2.4471 | 0.1069 | |
| 3 | Foreign nationals from Kansai Airport | Foreign guests to Kyoto City | 5.0197 | 0.01438 | Yes |

Source: Author.

The Granger causality test can be applied to the relationship between foreign nationals from Kansai International Airport and the number of tourists to Kyoto Prefecture. First, the null hypothesis of Granger causality is that foreign nationals from Kansai International Airport do not Granger-cause the number of tourists to Kyoto Prefecture and the *p*-value of 0.01969 was considered statistically significant at n = 3 in Equation (3).

Second, the null hypothesis of Granger causality is that foreign nationals from Kansai International Airport do not Granger-cause the number of tourists to Kyoto Station and the *p*-value of 0.01865 was considered statistically significant at n = 3.

Third, the null hypothesis of Granger causality is that the number of passengers at Kyoto Station do not Granger-cause the number of tourist visitors to Kyoto City and the *p*-value of 0.03552 was considered statistically significant at n = 1.

Fourth, the null hypothesis of Granger causality is that foreign nationals from Kansai International Airport do not Granger-cause foreign guests to Kyoto City and the *p*-value of 0.01438 was considered statistically significant at n = 3. The above results allow us to conclude that a reduction in transport costs induces an increase in tourists.

(3)　The Granger causality from the number of foreign tourists to revenues

The tests of Granger causality from the number of foreign tourists to revenues in the tourism industry in thirteen countries using cross-country data result in significant *p*-values. This means that the priority in sequencing the segment of foreign tourists and the segment of revenues in the tourism industry affects the result.

As shown in Table 6, the Granger causality test can be applied to the relationship between the number of foreign tourists and revenues in the tourism industry in cases in which the lag is 2 years. The null hypothesis of Granger causality is that revenues in the tourism industry does not Granger-cause the number of foreign tourists and the *p*-values of 0.05958, 0.05447, and 0.08539 were considered statistically significant in the cases of Spain, Mexico, and Turkey, respectively.

**Table 6.** Results of Granger causality tests (without constant).

| Country | Segment 1 | Segment 2 | Granger Causality Tests | | |
| --- | --- | --- | --- | --- | --- |
| | | | F-Test | *p*-Value | Significance |
| Spain | the number of tourists | the revenue of tourism | 1.9864 | 0.1537 | |
| Spain | the revenue of tourism | the number of tourists | 3.0843 | 0.05958 | Yes |
| Mexico | the number of tourists | the revenue of tourism | 0.10194 | 0.9034 | |
| Mexico | the revenue of tourism | the number of tourists | 3.1915 | 0.05447 | Yes |
| Turkey | the number of tourists | the revenue of tourism | 1.0518 | 0.3611 | |
| Turkey | the revenue of tourism | the number of tourists | 2.6598 | 0.08539 | Yes |
| Malaysia | the number of tourists | the revenue of tourism | 3.6833 | 0.03634 | Yes |
| Malaysia | the revenue of tourism | the number of tourists | 2.2743 | 0.1192 | |
| US | the number of tourists | the revenue of tourism | 2.6226 | 0.08816 | Yes |
| US | the revenue of tourism | the number of tourists | 1.4384 | 0.2522 | |
| China | the number of tourists | the revenue of tourism | 4.0377 | 0.02731 | Yes |
| China | the revenue of tourism | the number of tourists | 2.6798 | 0.08394 | |
| France | the number of tourists | the revenue of tourism | 1.51 | 0.2362 | |
| France | the revenue of tourism | the number of tourists | 0.37684 | 0.689 | |
| Italy | the number of tourists | the revenue of tourism | 0.98684 | 0.3838 | |
| Italy | the revenue of tourism | the number of tourists | 2.2377 | 0.1231 | |
| Thailand | the number of tourists | the revenue of tourism | 0.38784 | 0.6817 | |
| Thailand | the revenue of tourism | the number of tourists | 0.36452 | 0.6974 | |
| Japan | the number of tourists | the revenue of tourism | 0.91279 | 0.4116 | |
| Japan | the revenue of tourism | the number of tourists | 0.64271 | 0.5325 | |
| England | the number of tourists | the revenue of tourism | 1.1837 | 0.3192 | |
| England | the revenue of tourism | the number of tourists | 0.21244 | 0.8097 | |
| Germany | the number of tourists | the revenue of tourism | 0.78436 | 0.465 | |
| Germany | the revenue of tourism | the number of tourists | 0.36928 | 0.6941 | |
| Australia | the number of tourists | the revenue of tourism | 0.65048 | 0.5286 | |
| Australia | the revenue of tourism | the number of tourists | 1.8934 | 0.167 | |

Source: Author.

Again, as shown in Table 6, the Granger causality test can also be applied to the relationship between the number of foreign tourists and revenues in the tourism industry. The null hypothesis of Granger causality is that the number of foreign tourists does not Granger-cause revenues in the tourism industry and the *p*-values of 0.03634, 0.08816, and 0.02731 were considered statistically significant in the cases of Malaysia, the United States of America, and China, respectively. In summary, Granger causality from the number of foreign tourists to revenues in the tourism industry was found for three countries.

## 5. Summary and Conclusions

The flowchart approach provides an architectural theory for industrial agglomeration. The sequence of the formation of segments; i.e., the iterative order, is the dynamic process through which an industrial agglomeration is built. The economies of sequence are defined as the selection and efficient building of any two segments from among the total number of segments of an industrial agglomeration. The integration of all the economies of sequence through their integration into a flowchart approach

provides the overall process of building an industrial agglomeration. The fact-finding on the 'economies of sequence' can be summarized as follows:

First, foreign nationals from Kansai International Airport Granger-cause the number of tourists to Kyoto Prefecture, foreign nationals from Kansai International Airport Granger-cause the number of foreign tourists to Kyoto City, and passengers at JR Kyoto Station Granger-cause the number of tourists to Kyoto City. Second, the number of foreign tourists Granger-causes revenues in the tourism industry in the cases of Malaysia, the United States of America, and China.

The first and second facts mean that segments related to the reduction in transport costs such as airports and railways in the segments of transportation infrastructure increased tourist numbers, resulting in an increase in revenues in the tourism industry.

Third, based on the dummy variable method, the Kyoto case study of a tourism industry agglomeration indicates that the promotion of the tourism industry to redevelop Kyoto station mall in 1997, the "Kyoto Winter Special" campaign in 2003, and the global "Travel & Leisure" destination SNS promotion in 2015 were effective in building the segments of a tourist agglomeration in Kyoto in 2001, 2008, and 2015, respectively.

In conclusion, based on the economies of sequence, the efficient sequence of segment construction is as follows: the first priority to reduce transport costs for tourists consist of the renovation of Kyoto Station and new express schedules that directly link Kansai Airport and Kyoto Station, with the subsequent segment to enhance the level of culture being a branding strategy through the promotion of the tourist industry. Lastly, segments to solve problems of 'overtourism' needed in the case of the Kyoto agglomeration require further discussion in the future.

The policy implications of this paper are as follows: The policy makers need to take into account 'economies of sequence' in the dynamic processes of building the segments of an agglomeration. The results of Granger causality tests of the economies of sequence show that there exist the causality relationships in the dynamic processes. Accordingly, the completion of agglomeration construction depends on whether the judgment of the policy makers allows correct sequencing of the segments in order to efficiently build them. It is necessary to identify the current stage of a region along the flowchart approach in order to consider the economies of sequence in building the segments of a tourism agglomeration. In the tourism industry agglomeration, policy makers specify the segments of an agglomeration, then sequence them according to economies of sequence obtained in the paper.

However, this paper analyzed the example of Kyoto as a case study and proposed a prototype model to apply to other cases in other regions, both in Japan and around the world. The agglomerations of manufacturing industries, such as the electric/electronics industry and the automobile industry, were analyzed by Kuchiki and Tsuji (2008) and so on. This paper applied the flowchart approach to build a tourism industry agglomeration. However, the flowchart approach is yet to be analyzed in other industries such as the information communication technology industry. In future we need to find new segments, explore the substitute data of segments, examine economies of sequence, and further establish the flowchart approach by increasing the number of samples.

**Funding:** This work was supported by Japan Society for the Promotion of Science (JSPS) Grant Number 17H04549.

**Acknowledgments:** We would like to thank Akemi Baba, Hideyoshi Sakai, Katsumi Nakayama and the referees for their valuable comments.

**Conflicts of Interest:** The author declares no conflict of interest.

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
