# Peer review of "On ‘Economies of Sequence’ in the Architectural Theory of Agglomeration: A Case of the Kyoto Tourism Industry"

_economies, doi:10.3390/economies8010015_

Round 1

Reviewer 1 Report

1)The authors used Granger causality test without pre-tests (stationarity, cointegration…).

2)I cannot find abstract

3)Furthermore, the paper can be benefited from the following studies:

Songling, Y., Ishtiaq, M., & Thanh, B. T. (2019). Tourism industry and economic growth Nexus in Beijing, China. Economies, 7(1), 25.

Giap, T., Gopalan, S., & Ye, Y. (2016). Drivers of growth in the travel and tourism industry in Malaysia: A Geweke causality analysis. Economies, 4(1), 3.

Domínguez, A., & Mendez, C. (2019). Industrial Productivity Divergence and Input-Output Network Structures: Evidence from Japan 1973–2012. Economies, 7(2), 52.

Tzeremes, P. (2019). Does the Environmental Kuznets Curve Exist in the Chinese Regions?. Global Economic Review, 48(4), 363-377.

Reviewer 2 Report

Referee report on “On ‘Economies of Sequence’ in the Architectural Theory of Agglomeration: A Case of the Kyoto Tourism Industry”

General Preview:

This paper attempts at studying the economies of sequence as being the efficient sequence of segment construction. They show that:

The first priority to reduce transportation costs for tourists is the renovation of Kyoto Station and new express schedules that directly link Kansai Airport and Kyoto Station, with the subsequent segment to enhance the level of culture being a branding strategy through the promotion of the tourist industry. Segments to solve problems of ‘overtourism’ are needed in the case of the Kyoto agglomeration for discussion of future work. It is necessary to identify the current stage of a region along with the flowchart approach in order to consider the economies of sequence in building the segments of a tourism agglomeration.

The paper’s contribution is somehow interesting. However, the paper requires a number of improvements to be published.

For the ease of the follow up by the authors, I have decided to list the remarks on a section-by-section basis. Having said that, find below a list of suggested improvements:

Introduction: The authors did not provide an abstract at the beginning of the paper. Accordingly, it is highly recommended that the authors include an abstract that briefly discusses the aim of the paper, the main findings and the contribution to the policy-making procedure. The introductory statement is vague. It is recommended that the authors show why countries are facing serious issues in terms of regional development The authors committed a logical fallacy of “Hasty Generalization” in the phrase “Every country” at the beginning of the paper. Hence, the authors are advised to be cautious in choosing their terminology. IN line #17-19 the authors mention that Fujita, Krugman, and Venables (1999) developed a new theory of economic geography. What is this new theory? And how does it relate to the studied topic by the authors of this paper? A similar issue is observed in lines #22-23, where the authors argue that Porter (1990) developed a diamond model to specify the “four factors contributing to the geographical management theory”. What are these factors? And how they relate to the topic of economies of sequence? In line #31 the authors provide a definition about the Economies of Sequence that needs to be referenced. This section is titled “Introduction” but does not serve either the purpose of an introduction nor a literature review as it: Lacks a motivation behind the topic Lacks enough literature investigation into previous studies that spoke about the economies of sequence, tourism or growth. The authors are highly advised to either extend the “Introduction” to account for previous studies on the topic to create a new section for the literature review.

The authors can resort to the below work for additional references:

Dibeh, G., Fakih, A., & Marrouch, W. (2019). Tourism–growth nexus under duress: Lebanon during the Syrian crisis. Tourism Economics, 1354816619836338. Fakih, A., & Marrouch, W. (2015). The electricity consumption, employment and growth nexus: evidence from Lebanon. OPEC Energy Review39(3), 298-321. Sala-i-Martin, X. (1994). Regional cohesion: evidence and theories of regional growth and convergence. Abosedra, S., & Fakih, A. (2014). The relationships between economic growth, financial deepening, and information and communication technology: Empirical evidence from Lebanon. Journal of Economic Research19(1), 1-17. Tiebout, C. M. (1956). Exports and regional economic growth. Journal of political economy64(2), 160-164. Lacks a proper organization of ideas that are presented as bit and bytes here and there. The Flowchart Approach and Economies of Sequence Sentences “The model shows -> periphery” in lines 62-63 and “Projects aimed -> Special” campaign” in line 330-332 are not understandable and needs revision/further elaboration. This section – and the paper in general - includes some typos and grammatical mistakes. Accordingly, consulting a proofreader for a final revision of the paper is recommended before publication. I list below some of these typos/grammatical mistakes: Line 99 page 4: “The flowchart approach in shown Figure 3” → “The flowchart approach is shown in Figure 3” Line 131 page 6: “s1, s2 and s3” → “s1, s2, and s3”. This is referred to as the Oxford comma. This is a commonly committed mistake that can be sometimes unintentional or due to the lack of knowledge on it is importance (The usage of the Oxford Comma is required under the APA standards and conventions)

Line 224 page 9: “vegetables, Uji tea, soybeans and azuki beans” → “vegetables, Uji tea, soybeans, and azuki beans”

Line 225 page 9: “vegetables, pickles and confectionary” → “vegetables, pickles, and confectionary”

Line 228 page 9: “in an historical backstreet” → “in a historical backstreet”

Line 250-251 page 9: “projects were set to attracts tourists from abroad” Line 258 page 10: “paper lanterns for a distance of five kilometers”

The development of the station mall and the strengthening of cultural elements in Kyoto This section sounds like an advertisement for tourists rather than a part of a research paper. Example: “Yatsuhashi is a famous Kyoto confectioner, and Nadaman is a famous Kyoto restaurant”. I believe that this can be included in the introduction – in a brief way – to be perhaps used as a motivation or to give a preview of why the researchers are interested in studying “Koyoto” At the moment, I am on page 9, i.e. I already finished reading more than half of the paper and I am still being told about the touristic places in Koyoto. This paper is intended to reach people in academia that are usually interested in the empirical work and the findings of the paper rather than a touristic guide for the various places in Koyoto.

The dummy variables method and Granger causality tests for identifying the sequence of segments: The above title should be shortened to something more straight forward such as “Methodology and Empirical Results”

Summary and Conclusions: Authors are encouraged to elaborate more on the policy implications of their paper, acknowledge some limitations of their model/data used (if any), and to set up some ideas for future research investigations.

Overall Evaluation: the paper requires major revision to be accepted for publication.

Round 2

Reviewer 2 Report

The authors addressed well the previous suggestions. I have no additional comments.

This manuscript is a resubmission of an earlier submission. The following is a list of the peer review reports and author responses from that submission.

Round 1

Reviewer 1 Report

I am very sorry but manuscrypt does not meet the requirements of scientific work. Until they are removed, I will not comment on its scientific value. I suggest that authors learn about the requirements for articles and improvement. There are many inaccuracies and ambiguities. This applies both to the content of the work and the lack of literature, and ending with the tables and the quality of the drawings. In addition, some ambiguities make it impossible to assess the article, eg how to refer to a particular pattern? No pattern has a numbering. The information about the article is also incompatible with the article itself (eg number of pages).

I suggest you read the texts already published in this magazine and format your text similarly.

It surprises me that in this article there is no question about the phenomenon, which is overtourism, important in this subject.

Reviewer 2 Report

This paper develops econometric methods to identify ‘economies of sequence’ and proposes an econometric method based on Granger causality tests and taking into account an examination of case studies on Kyoto in Japan and cross-country data from the United Nations World Tourism Organization. The use of a dummy variable method for analysis of the Kyoto case study of the tourism industry agglomeration indicated that the project to redevelop Kyoto station mall in 1997, the “Kyoto Winter Special” campaign in 2003, and the global ‘Travel & Leisure’ destination campaign in 2015 were effective in building the segments of a tourist agglomeration in Kyoto in 2001, 2008, and 2015, respectively. The analysis in this paper is comprehensive and detailed, I have the following comments for the authors to consider. However, the paper has several drawbacks.

Please keep the abstract as brief as possible. It would be good to take a closer look at your abstract to make it more accessible. Sometimes when abstract is hard to comprehend, an average reader avoids reading the paper. So it would be in your own interest to make the abstract simple with a clear mechanism including the last line of your abstract as a general forward-looking conclusion on this topic

The introduction should be better organised. It should discuss the main results and how it contributes to the closely related literature

The literature review is confusing. I think that the authors should specify that there are different approaches to handle non desirable factors under the strong and week disposability assumption and that among these the authors select their approach, justifying their option. What is the main novelty when contrasting this study with previous studies? Please reinforce the value added of the paper.

It is important to identify the most important result and explain how the result is obtained and why it matters. Please make sure that your main finding is robust and you are able to explain the mechanism of the impact with unambiguous supporting evidence in your paper.

Make clear the research questions of the paper. Use the research questions to form relevant hypotheses.

The methodology is at best casual, residing in the use of a mainstream approach. The econometric exercise lacks originality and complexity though at first glance appears to be correctly conducted. The authors used a simple OLS models without a preliminary analysis (heteroscedasticity, autocorrelation, multicollinearity, e.t.c.). Moreover, they said that employ Granger causality test, but which one? If they developed Granger causality test (as they said), using the F-test is not Granger causality test.

Results section is very briefly; I think that the analysis in this paper should be more comprehensive

The study policy implication seems rather scanty. I think the authors have to provide more economic rationale for their empirical results.

The conclusions should be further developed in order to explore the main advantages of the approach herein followed. What are the policy implications? What are the main differences regarding the outcomes obtained with this approach and the other approaches available in the scientific literature?

Future research agenda could be enriched.